# Clinical and whole genome characterization of SARS-CoV-2 in India

**Radhakrishna Muttineni[1]ᵒ, Nagamani Kammili[2]ᵒ, Thrilok Chander Bingi[3]ᵒ, Raja Rao M.[3]ᵒ, Kalyani Putty[4]ᵒ, Pankaj Singh Dholaniya****[5]ᵒ, Ravi Kumar Puli[6], Sunitha Pakalapati[2], Saritha S.[5], Shekar K.[1], Mallikarjuna Reddy Doodipala[7], Amit A. Upadhyay[8], Steven E. Bosinger[9], Rama R. Amara[8]\*, Anand K. Kondapi****[5]¤\***

**1** Virus Research Laboratory, Department of Zoology, Osmania University, Hyderabad, India, **2** Department of Microbiology, Gandhi Medical College and Hospital, Secunderabad, India, **3** Department of General Medicine, Gandhi Medical College and Hospital, Secunderabad, India, **4** Department of Veterinary Biotechnology, College of Veterinary Science, Rajendranagar, PVNR Telangana Veterinary University, Hyderabad, India, **5** Department of Biotechnology and Bioinformatics, School of Life Sciences, University of Hyderabad, Hyderabad, India, **6** Telangana State Council for Science and Technology, Government of Telangana, Hyderabad, India, **7** Department of Mathematics, GITAM University, Hyderabad, India, **8** Department of Microbiology and Immunology, Emory Vaccine Center, Yerkes National Primate Research Center, Emory University, Atlanta, GA, United States of America, **9** Department of Pathology and Laboratory Medicine, Emory Vaccine Center, Yerkes National Primate Research Center, Emory University, Atlanta, GA, United States of America

ᵒ These authors contributed equally to this work.
¤ Current address: Department of Microbiology, Immunology, and Pathology, Colorado State University, Fort Collins, CO, United States of America
\* akondapi@gmail.com (AKK); ramara@emory.edu (RRA)

**Data Availability Statement:** All relevant data are within the paper and its Supporting Information file.

**Funding:** The authors received no specific funding for this work.

## Abstract

We report clinical profile of hundred and nine patients with SARS CoV-2 infection, and whole genome sequences (WGS) of seven virus isolates from the first reported cases in India, with various international travel histories. Comorbidities such as diabetes, hypertension, and cardiovascular disease were frequently associated with severity of the disease. WBC and neutrophil counts showed an increase, while lymphocyte counts decreased in patients with severe infection suggesting a possible neutrophil mediated organ damage, while immune activity may be diminished with decrease in lymphocytes leading to disease severity. Increase in SGOT, SGPT and blood urea suggests the functional deficiencies of liver, heart, and kidney in patients who succumbed to the disease when compared to the group of recovered patients. The WGS analysis showed that these isolates were classified into two clades: I/A3i, and A2a (four according to GISAID: O, L, GR, and GH). Further, WGS phylogeny and travel history together indicate possible transmission from Middle East and Europe. Three S protein variants: Wuhan reference, D614G, and Y28H were identified predicted to possess different binding affinities to host ACE2.

## Introduction

The coronavirus disease 2019 (COVID-19) caused by the Severe Acute Respiratory Syndrome Coronavirus 2 (SARS-CoV-2) was first reported in December 2019 from Wuhan, China [1].

**Competing interests:** The authors have declared that no competing interests

Coronaviruses are enveloped positive sense RNA viruses ranging from 60 nm to 140 nm in diameter with a genome size of approximately 26–32 kb [2]. They possess a crown like structure formed by glycoprotein Spike (S) protein on the virus facilitating its recognition to surface receptor, hence the name coronavirus [3]. In the 1960s, human coronaviruses (HCoVs) were first described in patients with common cold. Since then, seven HCoVs have been known to infect humans: 229E, OC43, SARS-CoV, NL63, HKU1, MERS-CoV (Middle East respiratory syndrome corona virus), and SARS-CoV-2 [4–6]. All the HCoVs are zoonotic with bats as key reservoirs of the virus [7, 8]. Over the last two decades, the world has experienced three different outbreaks caused by corona viruses with extremely high morbidity rates: SARS-CoV in 2002, MERS-CoV in 2012, and COVID-19 in 2019 [9, 10]. SARS-CoV-2 virus shows high pathogenicity of human transmission than MERS-CoV and SARS-CoV [11, 12].

With its complex disease morbidity and mortality, COVID-19 has raised global concern and is characterized as a pandemic event by the World Health Organization (WHO) on March 11, 2020. As reported by WHO, globally, as of 11 December, 2020, there have been 69,143,017 confirmed cases of COVID-19, including 1,576,516 deaths. Infections in India were reported from March 2020; as per the report by WHO there have been 9,796,769 confirmed cases of COVID-19 with 142,186 deaths in India as on 11th December 2020 [13].

Molecular features of the virus could be realized by undertesting genotypic and phenotypic properties of the viruses. Whole genome sequence (WGS) data is an excellent resource in understanding the evolution of the virus, assist in tracing pathways promoting infection, etc; understanding of which can assist in the development of diagnostics, therapeutic and preventive strategies. The SARS CoV-2 genomic resource GISAID database (https://www.gisaid.org/) contained a compilation of 70,614 SARS-CoV-2 complete genomes contributed by researchers from across the world by 23rd July 2020. Among these, 305 complete genome sequences of Indian SARS-CoV-2 isolates were archived on GISAID. Phylogenetic analysis revealed clustering of Indian isolates with existing global sequences of SARS-CoV-2 in two separate clusters indicating two different introductions of the virus genotype into the country [14]. Furthermore, the virus exhibited 96.09% nucleotide identity with Bat CoV RaTG13 sequence, though phenotypically they exhibit distinct features [14]. For a comprehensive understanding of SARS-CoV-2 evolution, including different evolutionary strategies adapted by the virus, continuous monitoring of WGS of the virus strains from different parts of the world is crucial. The primary objective of the current study is to present clinical profile of hundred and nine SARS-CoV-2 patients admitted to a COVID-19 referral Gandhi hospital, Hyderabad, Telangana State, India of which fifty-seven patients recovered and fifty-two patients succumbed to the illness. Secondary objectives are to: a) perform comprehensive WGS analysis of seven strains of SARS-CoV-2 from among the first reported (during February-March, 2020) SARS-CoV-2 cases in Hyderabad, India with international travel history from Europe, USA, Indonesia, and United Arab Emirates. These patients were screened for COVID-19 symptoms at Rajiv Gandhi International airport, Hyderabad, India, quarantined, and were tested positive for COVID-19 by RT-PCR. These patients were admitted in the COVID referral hospital in Hyderabad and enrolled in the study. b) identify changes in the structure of spike protein among the isolates, and c) quantify differential expression of host ACE2 in the seven different cases.

## Materials and methods

### Ethical approval

This study was approved by Institutional ethics committee, and was carried out as per the Institutional ethics committee approval No. IEC/GMC/2020/02/40 dated 04/04/2020 of

Gandhi Medical College and Hospital. In the context of emerging infectious disease conditions, the requirement for written informed consent was waived.

## Clinical specimen collection and detection of SARS-CoV-2 nucleic acid

Clinical specimens for SARS-CoV-2 diagnostic testing were obtained in accordance with WHO guidelines [13]. Nasopharyngeal and oropharyngeal swab specimens were collected with synthetic fiber swabs; each swab was inserted into a separate sterile tube containing 2 to 3 ml of viral transport medium (QIAGEN). RNA extraction was preformed using QIAamp Viral RNA Mini Kit (QIAGEN, Cat#52906) as per the manufacturer's instructions. The extracted RNA was immediately used for testing the presence of SARS-CoV-2 nucleic acid using the real time RT-PCR protocol [14] recommended by National institute of virology, Pune, India, using Super Script™ III Platinum® One-Step Quantitative Kit (Invitrogen, Cat. No.11732088) [15].

## Clinical records

The medical records of the SARS-CoV-2 positive patients were analysed by the research team, Department of General Medicine, COVID-19 referral Gandhi Hospital and Medical College, Hyderabad, India. Physicians and researchers reviewed the clinical, laboratory, radiological characteristics, and treatment measures of the patients.

## Whole genome sequencing

The WGS was carried out at Med Genome Labs Ltd., Bangalore, India. NEB Ultra II directional RNA-Seq Library Prep kit (NEB, Cat# E7760L) was used to prepare libraries for total RNA sequencing. Libraries were pooled and diluted to final optimal loading concentrations for cluster amplification on Illumina flow cell followed by sequencing on Illumina HiSeq X instrument to generate 150bp paired end reads. The quality of the reads was evaluated using Fast QC v0.11.9 [16] and cut adapt v2.9 [17] was used to trim adapters and remove contaminants. The reads were aligned to the human reference genome hg19 using STAR v2.4 [18]. The reads that do not align to the human reference were aligned to the Wuhan reference genome downloaded from NCBI (Ref Seq NC_045512.2) using BWA v.0.7.12 aligner [19]. The reads mapping to the genes were counted using bed tools v2.26.0 [20]. The aligned reads were sorted, and then variant calling was performed using GATK variant caller v 4.1.0.0 [21]. The variants identified were then annotated to the genes. The variant class, amino acid changes and other relevant annotations were added to the variants. The variants were then compared with the genomes available in NCBI and GISAID database.

## Phylogeny reconstruction

Phylogenetic analysis of the WGS data of samples was performed by multiple sequence alignment of the sequences with the genomes available in NCBI and GISAID database. The phylogenetic tree in *newick* file format was downloaded from Indian Coronavirus Genome Datasets maintained by CSIR Institute of Genomics and Integrative Biology (CSIR-IGIB; http://clingen. igib.res.in/genepi/phylovis/) for Indian SARS-CoV-2 sequences and visualized using FigTree v1.4.4. Further, BLAST was performed for each sequence of the current study on the GISAID database against all the SARS-CoV-2 genome sequences. Top 30 sequence hits for each isolate were downloaded and merged into a single file. This resulted in total of 142 sequences which also included the seven sequences of the current study. The sequence alignment was performed using MUSCLE program available in MEGA v10.1.7, and the phylogenetic tree was

reconstructed using Neighbor-joining method and Kimura 2 parameter as nucleotide substitution model with 1000 bootstrap values. For phylogenetic analysis of the S protein, protein sequences of SARS-CoV-2 S protein of all Indian isolates were downloaded from GISAID database. The multiple sequence alignment was performed using MUSCLE algorithm in MEGA v10.1.7. The phylogenetic tree was reconstructed using Neighbor-joining method with 1000 bootstrap validation and p-distance substitution model.

## Structural analysis of spike protein

Homology modelling based on the spike sequence was carried out using Swiss Model [22]. Obtained model was further optimized using fragment guided molecular dynamics simulation [23]. Optimized model was visualized in Pymol. Trimeric spike protein optimized above was docked using protein-protein docking using ClusPro Srver; highest neigbours were taken as most probable conformation. Docked conformation was visualized in Pymol [24–26]. Binding constant & energy, and dissociation constant values were computed at Web server PRODIGY [27].

## Statistical analysis

In the current study, categorical variables were described as frequency rates and percentages; continuous variables were described as mean, median, and interquartile range. Means for continuous variables were compared using independent two sample t- test statistics (for two-tail) when the data were normally distributed. On the other hand, categorical type variables were investigated using a non-parametric Mann-Whitney-U-test. For determining the proportions for categorical variables, a non-parametric Chi-square test was used, then complete data was cross tabulated (rows v/s columns) and analyzed. The Fisher exact test was used when the data were limited. In each case, a two-sided $\alpha$ of less than 0·05 was considered as statistically significant. The statistical analyses were done using SPSS ver. 20 software.

## Results and discussion

### Clinical, radiological, functional, and immunological profile of COVID-19 patients

The current study enrolled patients (N = 109) diagnosed for SARS-CoV-2 during February 2020 to May 2020, admitted to a tertiary referral hospital in Hyderabad, India. Patients exhibiting hypoxia, shortness of breath, altered sensorium, haemoptysis, acute kidney injury (AKI), chronic kidney disease (CKD), aphasia, chest pain, dysphagia, oral ulcers, hypoxia, metabolic encephalopathy, renal failure, tracheostomy care, and sudden syncope were admitted to intensive care unit (ICU). Majority of the patients had frequently reported fever (37%) and cough (39%). Patients with comorbidities such as diabetes (33%), hypertension (31%), CKD (7%), and cardio vascular disease (CVD) (9%) became severely ill, even when intervened through respiratory support along with oxygen, and medical support with antibacterials, antivirals, hydroxy chloroquine (HCQ), and steroids. Conditions such as cancer (1%), septic shock (11%), acute respiratory distress syndrome (ARDS) (40%), CKD (5%), non-ICU sudden collapse (1%), were seen in patients that could not be recovered and succumbed to the disease (Table 1). For clinical comparison in the current study, the patients were grouped in to recovered (R) (N = 57; male (n): 37, female (n): 20, aged 3 years-72 years) and succumbed (S) (N = 52; male (n): 41, female (n):11, aged 2 months-85 years). Median age of recovered patients was found to be 32 years (26y-35y), and death patients to be 61 years (35y-85y). Between R and S groups there was a significant difference (P<0.05) in favour of S group in the onset of

**Table 1. Comparison of clinical findings among Recovered (R) and Succumbed group (S).**

| Parameter | Number | Recovered number (R) | Succumbed number (S) | P value |
|---|---|---|---|---|
| Number of patients | 109(100%) | 57(52%) | 52 (48%) | - |
| Median age in years (IQR) | 46(41–50) | 32(26–35) | 61(35–85) | <0.0001* |
| Sex (Male, Female median age in years) | 51, 37 | 46, 27 | 61, 47 | <0.0001* |
| Male | 78(71%) | 37(64%) | 41(78%) | 0.02* |
| Female | 31(28%) | 20(35%) | 11(21%) | 0.02* |
| **Signs & Symptoms** | | | | |
| Fever | 40(37%) | 8(14%) | 32(62%) | 0.03* |
| Cough | 42(39%) | 10(17%) | 32(62%) | <0.0001* |
| Sore throat | 8(7%) | 4(7%) | 4(7%) | 0.36 |
| Headache | 6(6%) | 3(5%) | 3(5%) | 0.28 |
| Cold/Loss of Smell/Sneezing | 1(1%) | 0 | 1(1%) | 0.9 |
| Breathing difficulty | 46(42%) | 2(3%) | 44(84%) | <0.001* |
| Respiratory Rate>24 breaths per min | 83(76%) | 51 (89%) | 32(61) | <0.001* |
| SP O2 (%) | 85(78) | 57(100) | 28(54) | <0.001* |
| Retractions | 3(2%) | 0 | 3(5%) | 0.31 |
| Crepts | 23(21%) | 0 | 23(44%) | 0.02* |
| Diarrhoea | 5(4%) | 0 | 5(9%) | 0.13 |
| **Comorbidities** | | | | |
| Diabetes | 37(33%) | 5(8%) | 32(61%) | <0.001* |
| <5 | 12(11%) | 5(8%) | 7(13%) | <0.001* |
| 5–25 | 25(22%) | 0 | 25(48%) | <0.001* |
| Hypertension | 34(31%) | 3(5%) | 31(60%) | <0.001* |
| <5 | 12(11%) | 3(5%) | 9(17%) | <0.001* |
| 5–25 | 22(20%) | 0 | 22(42%) | <0.001* |
| Systolic Blood Pressure Median (IQR) | 120(110–130) | 120 (110–128) | 131(110–150) | <0.0001* |
| Diastolic Blood Pressure median (IQR) | 85(80–92) | 80 (80–90) | 79(60–90) | <0.0001* |
| CVD (Cardio Vascular Disease) (yrs) | 10(9%) | 1(1%) | 9(17%) | 0.04* |
| <5 | 8(7%) | 1(1%) | 7(13%) | 0.04* |
| >5 | 2(1%) | 0 | 2(3%) | 0.04* |
| CKD (Chronic Kidney Disease) | 8(7%) | 0 | 8(15%) | 0.99 |
| Hypothyroidism | 5(5%) | 0 | 5(10%) | 0.99 |
| Asthma | 5(4%) | 1(1%) | 4(7%) | 0.01* |
| **ECG** | | | | |
| Normal sinus rhythm | 55(50%) | 54(94%) | 1(1%) | <0.001* |
| Sinus tachy | 19(17%) | 1(1%) | 18(35%) | <0.001* |
| Median Heart Rate/PR bpm (IQR) | 80(75–110) | 86(78–95) | 102(80–123) | <0.001* |
| **RNA CT Values (median, IQR)** | | | | |
| E | 18(15–25) | 20(18–22) | 20(17–23) | <0.04* |
| RdRP | 26(20–35) | 28(23–32) | 24(21–26) | <0.01* |
| ORF | 24(20–31) | 26(24–28) | 23(21–25) | <0.01* |
| **Date of illness to Recovery/Death** | | | | |
| Date of illness to recovery (median days)/Death Days (IQR) | 15(10–20) | 16(8–20) | 9(6–12) | 0.04* |
| Length of hospital stay in days | 13(11–21) | 16(7–20) | 9(4–12) | 0.04* |
| **Radiological Findings** | | | | |
| Normal | 44(40%) | 39(68%) | 5(9%) | <0.001* |
| Infiltrations | 19(17%) | 1(1%) | 18(35%) | <0.001* |
| Haziness | 7(6%) | 3(5%) | 4(7%) | <0.001* |

*(Continued)*

**Table 1.** (Continued)

| Parameter | Number | Recovered number (R) | Succumbed number (S) | P value |
|---|---|---|---|---|
| GGO | 7(6%) | 1(1%) | 6(11%) | <0.001* |
| Consolidation | 4(3%) | 2(3%) | 2(3%) | <0.001* |
| **Treatment** | | | | |
| Antibiotic | 53(49%) | 11(19%) | 42(80%) | <0.001* |
| Antiviral | 51(47%) | 9(16%) | 42(80%) | <0.001* |
| Hydroxy Chloroquine | 18(16%) | 1(1%) | 17(28%) | <0.05* |
| Steroid | 4(3%) | 0(0) | 4(7%) | <0.001* |
| **Cause of Death** | | | | |
| ARDS (Acute respiratory distress syndrome) | 21(19%) | 0 | 21(40%) | 0.49 |
| CKD (Chronic kidney disease) | 3(3%) | 0 | 3(5%) | 0.50 |
| Septic Shock | 6(5%) | 0 | 6(11%) | 0.56 |
| Non-ICU Sudden Death | 1(1%) | 0 | 1(1%) | - |
| Cancer | 1(1%) | 0 | 1(1%) | - |
| Meningoencephalitis | 1(1%) | 0 | 1(1%) | - |

Numbers in the brackets indicate percentage of patients enrolled in the current study that are exhibiting the respective parameters; unless mentioned otherwise.

*Indicates significant difference.

certain symptoms (fever, cough, breathing difficulty, and crepts) and comorbidities (diabetes, hypertension, systolic blood pressure, diastolic blood pressure, and asthma) (Table 1). Significant difference (P<0.05) between these two groups (in favour of S group) was also observed in ECG, viral Ct values, majority of the pathological lung radiological findings (Table 1, Fig 1). Interestingly, significant difference in favor of S group was noticed with supportive therapy. Supportive therapy was provided with antibiotics (cefotaxime, azithromycin), antivirals (oseltamivir and lopinavir/ritonavir), HCQ, and steroids. This suggests that the treatment started in severely ill patients had no effect in the recovery of the patients. This also suggests that supportive therapy although statistically found significant, has no effective role in disease outcome.

Cause of death was significant (P<0.05) due to ARDS, CKD, and septic shock. Other non-significant causes of death include meningoencephalitis, cancer and non-ICU related sudden death (Table 1). It was shown previously that 90% of the deaths were due to ARDS followed by CKD (18%), and shock (12%) [28]. Previous studies reported that comorbidities, particularly the CVDs and chronic pulmonary diseases, were important to predict the in-hospital mortality in critically ill patients [29]. Our study reports that majority of patients in the S group had underlying diseases, especially hypertension, lung disease, and heart disease, more often with more than one comorbidity in individual patient. Sepsis was also a common complication found in this study, which might be directly caused by SARS-CoV-2 infection; further research is needed to investigate the pathogenesis of sepsis in COVID-19 illness.

In this study, radiological findings were obtained in only 40% of cases due to various logistic reasons. 35% of S group had infiltrates in two or more lobes, and almost 68% had normal radiological findings in R group (Fig 1). Overall, the findings commonly included patchy ground glass opacity (GGO) (11%) followed by confluent haziness (7%) in S group; in contrast consolidation was equal in both groups. Overall, our study highlights the radiological findings were consistent with the severity of infection varying from infiltrations through GGO to consolidations.

When immune response of R and S groups was compared, the results show a higher WBC and neutrophil count in the S group (Table 2). Neutrophilia is a hallmark of any acute

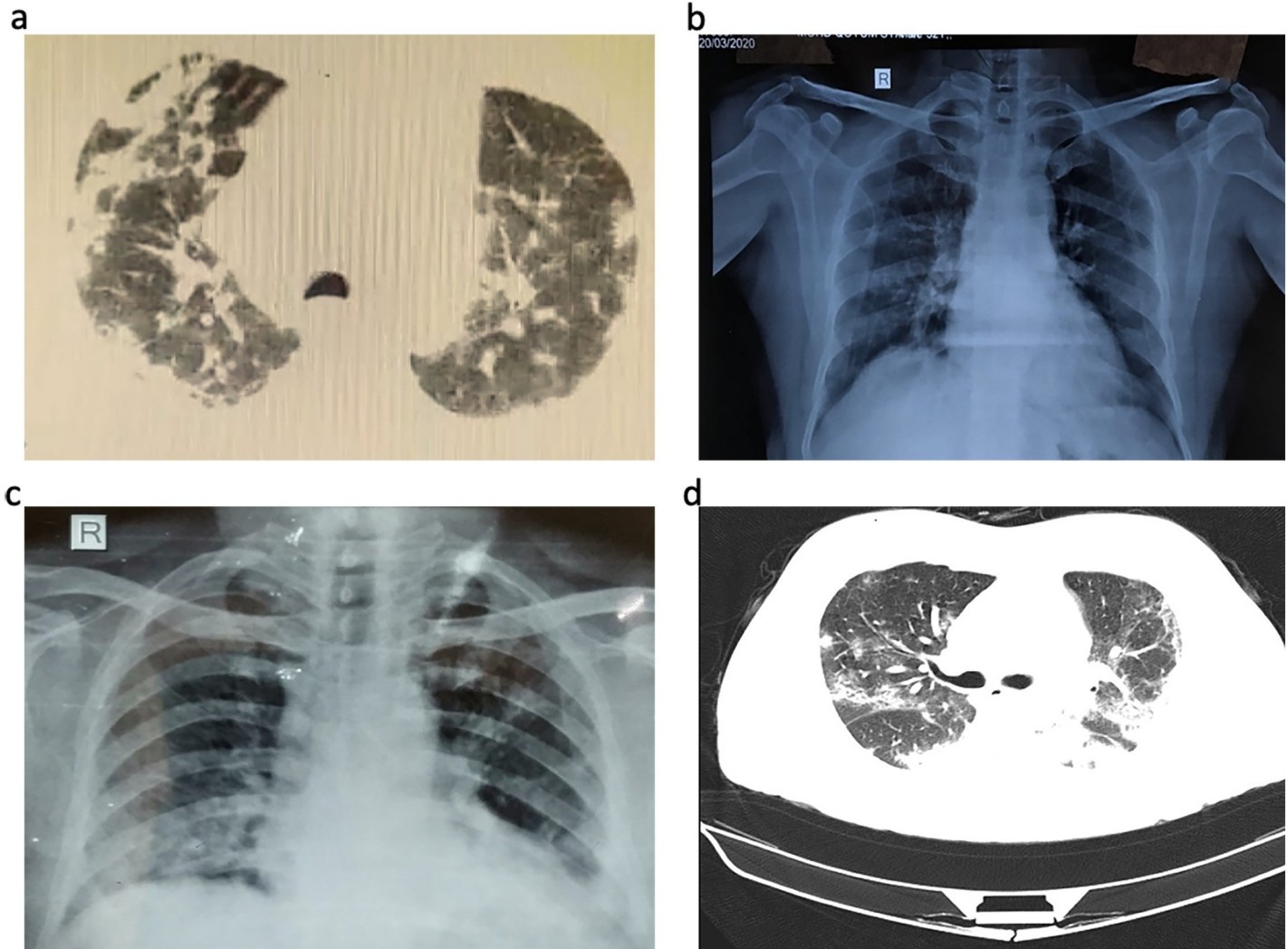

**Fig 1. Chest computed tomographic, and X-ray images. a)** Chest CT images of sample ID RK100 showing chest consolidation in central areas of RT upper lobe, RT middle lobe and bilateral basal areas **b)** Chest X ray of sample ID KN443 showing sub pleural ground glass opacities in bilateral mid and lower zones Lt>Rt haziness **c)** Chest X ray of sample ID RR1191showing bilateral lower zone and left retro cardiac consolidation, Lt>Rt haziness **d)** multifocal bilateral ground glass opacities and patchy consolidation in a patient suffering with COVID-19.

infection. It was also suggested before that aberrant activation of neutrophils contribute to the exacerbated host response in patients with severe COVID-19, predicting poor outcome [30]. Importantly neutrophil infiltration was also observed in pulmonary capillaries of COVID-19 patients leading to acute capillaritis with fibrin deposition, extravasation of neutrophils into the alveolar space, and neutrophilic mucositis [31]. In addition to this, the role of neutrophilia as a source of excess neutrophil extracellular traps (NETs) was also shown before [32]. NETs are web-like structures of DNA and proteins expelled from the neutrophils that capture and kill the pathogens [33]. In addition to being useful, excessive NET formation can trigger a cascade of inflammatory reactions that results in permanent organ damage to the pulmonary, cardiovascular, and renal systems, the most commonly affected organ systems in severe COVID-19 [6, 34–38].

Another interesting finding was that the lymphocyte count was low in S group compared to R group (Table 2); such a decrease in lymphocyte count was reported previously in pneumonia

**Table 2. Functional and immunological parameters of COVID-19 patients with variable severities.**

| Functional Parameter (SI Units) | Normal Range | All Patients(N = 109) Median (IQR) | Recovered (N = 57) | Succumbed (N = 52) | P Values |
|---|---|---|---|---|---|
| Total Serum Bilirubin, umol/L | 1.71–20.52 | 5.23 (4.99–15.91) | 5.385 (4.96–16.58) | 19.41 (18.82–40.5) | <0.05* |
| Serum Creatinine, umol/L | 61.88–114.9 | 64.5 (58.95–119.12) | 63.64 (40.66–86.62) | 92.33 (90.56–150.36) | <0.001* |
| BU mmol/L | 1.16–3.33 | 3.39 (1.21–5.57) | 2.11 (2.03–5.41) | 6.00 (5.68–15.34) | <0.001* |
| SGOT/AST, Units/Lit | 8.01–45 | 43.01 (43.55–95.13) | 13.43 (13.01–38.72) | 42.64 (40.00–90.5) | <0.001* |
| SGPT/ALT, Units/Lit | 7.01–56 | 19.05 (18.10–41.3) | 8.16 (5.56–27.08) | 21.92 (15.87–71.79) | <0.001* |
| AP, U/Lit | 44–147 | 117.5 (77.2–131.61) | 46.56 (42.56–133.7) | 48.38 (43.68–139.84) | 0.7 |
| Na, mmol/Lit | 136–145 | 138.00 (132.91–144.84) | 139.54 (137.57–144.51) | 140.6 (129.43–146.4) | <0.05* |
| K,mmol/Lit | 3.5–5.0 | 4.23 (3.45–5.01) | 4.16 (3.53–4.79) | 2.15 (3.36–5.24) | 0.42 |
| CL,mmol/Lit | 98–107 | 102.33 (96.81–107.85) | 102.36 (98.83–106.89) | 102.23 (95.1–109.54) | <0.001* |
| **Immunology** | **Normal Range** | **All Patients(N = 109) Median (IQR)** | **Recovered** | **Succumbed** | **P Values** |
| WBC X 10^9/L | 4.5–10 | 4.8(4.24–13.96) | 4.55 (4.35–12.47) | 6.37 (6.12–14.86) | <0.05* |
| Neutrophils, X10^9/L | 2.5–7.5 | 3.38 (2.56–10.95) | 2.40 (2.22–7.36) | 5.52 (4.43–13.63) | <0.05* |
| LymphocytesX10^9/L | 1–4.8 | 1.25 (0.7–4.15) | 1.66 (1.56–5.06) | 0.64 (0.39–2.17) | 0.52 |
| Monocytes, X10^9/L | 0.2–0.6 | 0.4 (0.32–0.96) | 0.38 (0.37–0.97) | 0.59 (0.45–1.68) | 0.16 |
| Eosinophiils, X10^9/L | 0.03–0.35 | 0.115 (0.07–0.53) | 0.12 (0.08–0.56) | 0.23 (0.21–0.97) | 0.88 |
| Platelets, X10^9/L | 150–450 | 250 (167.88–268.12) | 273 (250–450) | 263 (250–500) | 0.14 |

patients [39, 40]. This suggests that the increased lymphocytes in the R group may be contributing for the recovery of the patients, while lymphocytes may be damaged in the condition of severe infection condition, thus reducing recovery of S group. Lymphocytes play an important role in the maintenance of immune system function. After virus infection, alteration in total lymphocyte numbers varies with different viral pathogenic mechanisms [41]. Several mechanisms were shown before for virus induced lymphopenia including immune injuries from inflammatory mediators, virus attachment, exudation of circulating lymphocytes into inflammatory lung tissues [39]. Recent studies indicated a clear association of decrease in peripheral lymphocytes with the clinical characteristics of COVID-19 patients [39]. This indicates an impairment of the immune system during the course of SARS-CoV-2 infection. These alterations were also found in patients with pneumonia caused by MERS- CoV and SARS-CoV [42]. Marginal differences were seen in monocyte, eosinophil and platelet counts between R and S groups (Table 2).

When functional parameters were compared, large deviation of serum glutamic-oxaloacetic transaminase (SGOT), serum glutamic pyruvic transaminase (SGPT) and blood urea was seen when mean values were taken in consideration (Table 2). There was a significant increase in SGPT, SGOT and blood urea in S group compared to R group (Table 2). There was no significant change seen in alkaline phosphatase (AP), which indicates that liver may not be severely affected. Higher SGOT and SGPT levels could be due to effect of infection to both heart and liver [43]. Since blood urea was also significantly increased in S group, the observed increase may also be associated with comorbid diabetes in these patients (Table 2). In summary, neutrophil and lymphocyte count, the levels of SGOT and SGPT in COVID-19 patients may provide an indication on severity of the infection in Indian population infected with SARS CoV-2, while the comorbid conditions such as diabetes, hypertension and CVD would further enhance the complexity of the disease condition. Death was reported in the patients on an average of 9 (6–12) days of hospitalization, while recovery was noticed with in 16 (8–20) days of hospitalization (Table 1). A previous study reported number of days from hospital admission to death is 6.35+/-4.51 days [44].

Hospitalization was shorter in death patients, which is most likely due to rapid progression of disease in such severe cases. As per the recent health bulletin released by the Govt. of

Telangana, India, maximum number of deaths were reported in the age group of 51–60 (N = 52), followed by 61–70 (N = 48), 41–50 (N = 42), and 71–80 (N = 26) [45, 46]. Studies from China have reported similar findings that majority of the affected patients were middle aged and elderly people with median age of deceased patients being 69 years [28, 47]. It is evident that respiratory distress is the major cause of mortality. It is possible that ageing diminishes the lung function, regeneration, and remodelling thereby enhancing its susceptibility to the disease [48].

## Next generation sequencing

From the first reported cases in India that had international travel histories from USA, UK, Italy, Indonesia, United Arab Emirates during 20-02-2020 to 25-03-2020, n = 7 samples were sent to whole genome sequencing (n = 6 from R group and n = 1 from S group). > 4.9 Gb data was obtained by 151bp pair end sequencing. Of the total, more than 80% of bases have base quality > = Q30. The raw reads were pre-processed for adapter and contamination removal. Of the total raw reads, ~50–64% of the total reads were left after pre-processing. Out of total pre-processed reads, ~50–77% align to the host transcriptome/genome (human-hg19) using STAR aligner. The unaligned reads from host analysis were converted to fastq and then aligned to the SARS-CoV-2 reference genome (Genbank Ref seq NC_045512.2). A total of >200,000 reads mapped to the viral genome for all the samples, and of these 97% of the reads were properly paired. Coverage analysis of the genome showed 100% coverage for all the samples with an average read depth of >225X. Of the total bases of interest ~100% of the genome have more than 100X depth. The full-genome viral sequences were deposited in the dataset of GISAID (EPI_ISL_431101, EPI_ISL_431102, EPI_ISL_431103, EPI_ISL_431117, EPI_ISL_438139, EPI_ISL_437626, EPI_ISL_438138) and NCBI GenBank (MT415320, MT415321, MT415322, MT415323, MT477885, MT457402, MT457403).

## Genetic variation in the sequences

Mutations among the seven SARS-CoV-2 strains, were identified both at the genome level and corresponding amino acid level throughout the whole genome, with reference to the prototype SARS-CoV-2 Wuhan strain genome (NCBI Ref Seq NC_045512.2) (Table 3). Two synonymous (15324C>T in ORF1ab; 24130C>T in S) and two nonsynonymous mutations (21644T>C{Y28H} in the S gene; 29303C>T{P344S} in the N gene) were observed in the sample (OUMRK100/2020) collected from patient with travel history to Dubai. One synonymous (3037C>T in ORF1ab) and two nonsynonymous mutations (14408C>T{P4714L} in ORF1ab; 23403A>G{D614G} in the S gene) were observed in the sample (GMCKN318/2020) collected from patient with travel history to Italy. One synonymous (23929C>T in the S gene) and four nonsynonymous mutations (6312C>A{T2016K}, 11083G>T{L3606F}, 13730C>T{A4489V} in ORF1ab; 28311C>T{P13L} in the N gene) were observed in the sample (GMCKN443/2020) collected from patient with travel history to Indonesia. One synonymous (3037C>T in ORF1ab) and four nonsynonymous mutations (14408C>T{P4714L} in ORF1ab; 23403A>G {D614G} in the S gene; 28881G>A{R203L}, 28883G>C{G204R} in the N gene) were observed in the sample (GMCTC469/2020) collected from patient with travel history to UK. One synonymous (23929C>T in the S gene) and four nonsynonymous mutations (6312 C>A{T2016N}, 11083G>T{L3606F}, 13730C>T{A4489V} in ORF1ab; 28311C>T{P13L} in the N gene) were observed in the sample (OUMRK1090/2020) collected from patient with travel history to UAE. Two synonymous (3037C>T, 18877C>T in ORF1ab) and three nonsynonymous mutations (14408C>T{P4714L} in ORF1ab; 23403 A>G{D614G} in the S gene; 25563G>T{Q57H} in the N gene) were observed in the sample (GMCKP1125/2020) collected from patient with

**Table 3. Nucleotide polymorphisms and amino acid changes throughout the seven genomes.**

| | ORF1 ab | | | | | | | | | | | | | | Spike | | | | | | | | ORF3a | | Nucleocapsid | | | |
|---|---|---|---|---|---|---|---|---|---|---|---|---|---|---|---|---|---|---|---|---|---|---|---|---|---|---|---|---|
| | N | AA | N | AA | N | AA | N | AA | N | AA | N | AA | N | AA | N | AA | N | AA | N | AA | N | AA | N | AA | N | AA | N | AA |
| | 3037 | 925 | 6312 | 2,016 | 11,083 | 3,606 | 13,730 | 4,489 | 14,408 | 4,715 | 15,324 | 5,020 | 18,877 | 6205 | 21644 | 28 | 23403 | 614 | 23929 | 789 | 24130 | 856 | 25563 | 57 | 28311 | 13 | 29303 | 344 |
| **Wuhan Hu-1** | C | F | C | T | G | L | C | A | C | P | C | N | C | L | T | Y | A | D | C | Y | C | N | G | Q | C | P | C | P |
| OUMRK100/2020 | C | F | C | T | G | L | C | A | C | P | T | N | C | L | C | H | A | D | C | Y | T | N | G | Q | C | P | T | S |
| GMCKN318/2020 | T | F | C | T | G | L | C | A | T | L | C | N | C | L | T | Y | G | G | C | Y | C | N | G | Q | C | P | C | P |
| GMCKN443/2020 | C | F | C | T | T | F | T | V | C | P | C | N | C | L | T | Y | A | D | C | Y | C | N | G | Q | T | L | C | P |
| GMCTC469/2020 | T | F | C | T | G | L | C | A | T | L | C | N | C | L | T | Y | G | G | C | Y | C | N | G | Q | C | P | C | P |
| OUMRK1090/2020 | C | F | A | K | T | F | T | V | C | P | C | N | C | L | T | Y | A | D | C | Y | C | N | G | Q | T | L | C | P |
| GMCKP1125/2020 | T | F | C | T | G | L | C | A | T | L | C | N | T | L | T | Y | G | G | C | Y | C | N | T | H | C | P | C | P |
| GMC-RR1191/2020 | C | F | A | K | T | F | T | V | C | P | C | N | C | L | T | Y | A | D | C | Y | C | N | G | Q | T | L | C | P |

travel history to USA. One synonymous (23929C>T in the S gene) and four nonsynonymous mutations (6312 C>T{T2016K}, 11083G>T{L3606F}, 13730C>T{A4489V} in ORF1ab; 28311C>T{P13L} in the N gene) were observed in the sample (GMCRR1191/2020) collected from patient with cluster infection and no travel history, and who succumbed to the disease.

Orf1ab is a poly-protein, which, after translation, is cleaved into 15 nonstructural proteins (NSP). Total 4 nonsynonymous mutations were observed at 4 different positions among different samples. Mutation at position 6312 C>T{T2016K} is on **NSP3** in sample GMCKN443/2020, GMCRR1191/2020, and OUMRK1090/2020. NSP3 has 10 different conserved domains including the nucleic acid binding domain (cl24732: NAR superfamily). The mutation 6312 C>T{T2016K} falls on relative position of 1198 on NSP3 protein sequence. This position is related to the nucleic acid binding domain. The mutation 11083G>T{L3606F} was observed in NSP6 in sample GMCKN443/2020, GMCRR1191/2020, and OUMRK1090/2020. NSP6 is a 1945 amino acid long peptide containing putative transmembrane domain. In pp1ab it ranges from amino acid 3570 to 3895. Two mutations were observed in NSP11, in which the first mutation 13730:C>T{A4489V} was present in samples GMCKN443/2020 GMCRR1191/2020 and OUMRK1090/2020; and the second 14408:C>A{P4714L} was observed in samples GMCKN318/2020, GMCTC469/2020, and GMCKP1125/2020. NSP11 is RNA-dependent RNA polymerase (RdRp) having one conserved domain SARS-CoV-like RdRp (cd21591) (Fig 2).

N gene codes for nucleocapsid phosphoprotein (N Protein). Four different variants in N protein were reported among six out of seven different samples. The mutation 28311C>T {P13L}, was found in three samples GMCKN443/2020, OUMRK1090/2020, GMCRR1191/2020, while one mutation in each was observed in other three samples. The N protein of SARS-CoV-2 shares 90% sequence similarity with the N protein of SARS-CoV [49]. The N-protein is a RNA binding protein and has three intrinsically disordered regions (IDRs) from residues 1–44, 182–247 and 366–422 in case of SARS-CoV, which may modulate the RNA-binding activity [50, 51]. The two mutations out of four in N-protein reported from our analysis (28311C>T{P13L} and 28883G>C{G204R}) are present in IDRs suggesting there may be variation for RNA binding activity among the N-protein of different variants of SARS-CoV-2.

Several mutations were observed among the seven isolates with both synonymous and nonsynonymous mutations with pyrimidine exchanges (C to T or T to C) suggesting the

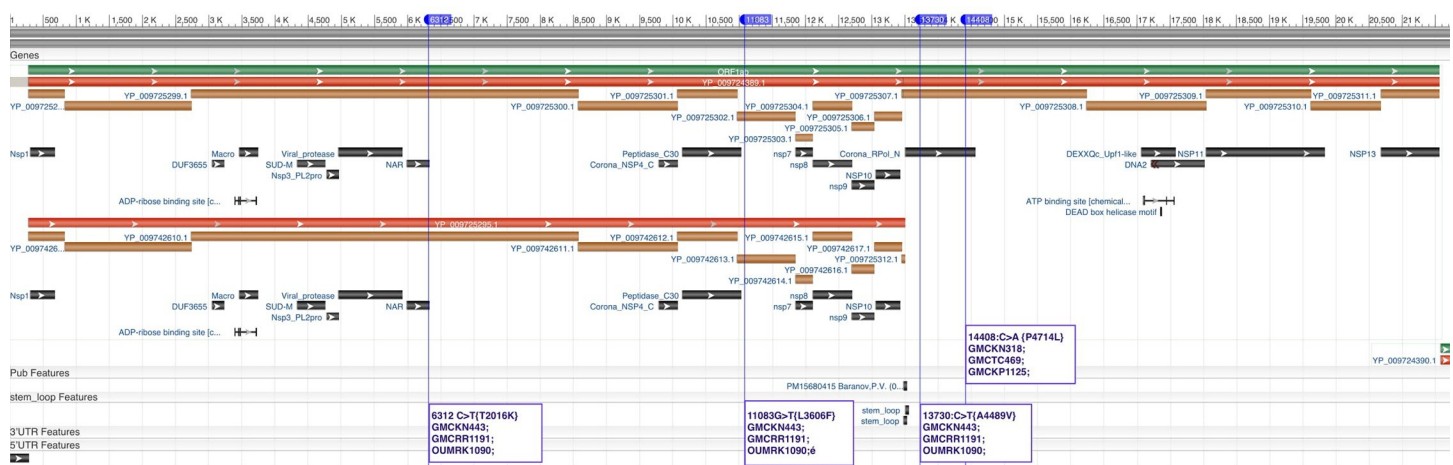

**Fig 2. Graphical representation of polyprotein synthesized by orf1ab (taken form NCBI sequence viewer).** The four different mutations observed in 6 different samples were mapped based on the position. Out of four, two mutations were observed in NSP3 and NSP6 each. The remaining two mutations were observed in RNA-dependent RNA polymerase (NSP11).

possibility of occurrence of transcription-induced mutations [52]. This could be due to high replication rates of the virus. Nonsynonymous mutations showed higher frequency than synonymous mutations in the ORF regions of the viral genomes suggesting that the selection of mutation may be biased towards viral phenotype changes which may be contributing to the observed variability in the severity of infection reported in different epidemiological studies [53].

### Phylogenetic analyses

The genomes sequenced in the current study were found to be in two clades, when analysed with the other available Indian SARS-CoV-2 sequences. Four of the seven sequences belong to clade-I/A3i; of which three sequences (OUMRK1090/2020, GMCRR1191/2020, and GMCKN443/2020) are clustered together, and one sequence (OUMRK100/2020) clustered separately in the same clade. The other three sequences (GMCTC469/2020, GMCKP1125/2020, and GMCKN318/2020) belong to the clade A2a in three different clusters (Fig 3). The sequences were further analyzed with other SARS-CoV-2 sequences worldwide (Fig 4). It can be seen that the virus sequence from patient with travel history to Dubai (OUMRK100/2020) clustered with sequences from UAE, Japan, and the original Wuhan-Hu-1 strain. Virus sequences from patients with travel history to Indonesia (GMCKN443/2020), UAE (OUMRK1090/2020), and no travel history (GMCRR1191/2020) clustered together along with two other Indian isolates, and a Singapore isolate. Sample GMCKP1125/2020 isolated from patient who travelled to India from USA via UAE can be seen closely clustered with one USA and three Saudi Arabian isolates. Virus sequence from patient with travel history to UK (GMCTC469/2020); and Italy (GMCKN318/2020) were found to be clustered with other England, Italian; and USA, Russian strains, respectively. The global comparison according to GISAID database revealed that three sequences from the current study: GMC-RK1090, GMC-RK1191, and GMC-KN443 are in the cluster labeled as clade O according the GISAID database. Other four sequences belonged to four different clades: L (GMC-RK100), G (GMC-KN318), GR (GMC-TC469) and GH (GMC-KP1125) (Fig 5).

These findings, suggest that there might be two distinct points of transmission: one is involving Middle East, and other Europe, this could be a transition point during travel for an airway transmission of the virus. A previous study suggested that divergence from a single point of Clade I/A3i suggests a single point of introduction into the country, and that the spread from this could be from a single outbreak [54]. The first sequence from this cluster in India was GMC-KN443 which was from a patient who travelled to India from Indonesia. However, the authors suggested that the strain is similar to those strains from Singapore and Philippines [54]. Furthermore, another study reported that isolate RK100 clustered with the ancestral first isolates from India that were originally reported from Kerala [55].

### Analysis of spike protein

Protein sequences for spike protein of SARS-CoV-2 of all Indian isolates were downloaded from GISAID database. Total 53 sequences were obtained in which 4 sequences (EPI_ISL_431101, EPI_ISL_431102, EPI_ISL_431103, and EPI_ISL_431117) were from Hyderabad isolates. The multiple sequence alignment was performed using MUSCLE algorithm in MEGA v10.1.7. The phylogenetic tree was constructed using Neighbor-joining method with 1000 bootstrap validation and p-distance substitution model. The results show four different genotypic variants (Fig 6). Among these, three amino acid variations were observed including reference Wuhan isolate. RK100 showed Y28H mutation, while sample nos KN318, TC469, and KP1125 showed D614G mutation, and samples KN443, RK1090 and

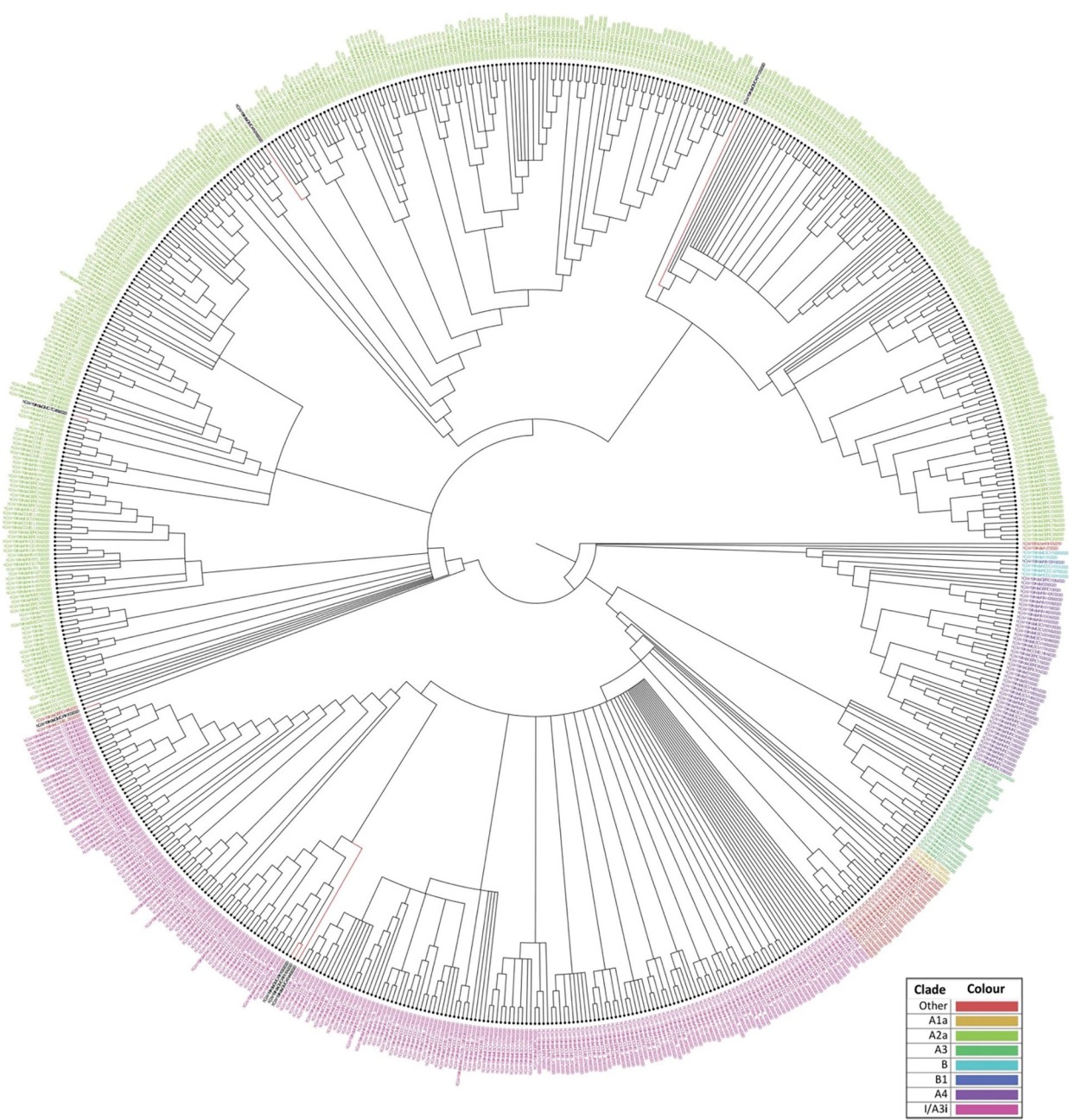

**Fig 3. Phylogenetic tree of Indian SARS-CoV-2 sequences according to the dataset available at CSIR Institute of Genomics and Integrative Biology.** The sequences form the current study are labeled in black font. Sequences OUMRK1090/2020, GMCRR1191/2020, GMCKN443/2020, and OUMRK100/2020 are in clade I/A3i. Sequences GMCTC469/2020, GMCKP1125/2020, and GMCKN318/202 are clustered in clade A2a.

RR1191 spike protein are similar to Wuhan reference. Thus, among seven samples there are three phenotypic variants of spike protein.

Total 53 different types of proteins were identified in GISAID database to date. Among seven Hyderabad isolated total 4 different types of Spike proteins are observed (Red).

Structure prediction analysis of spike protein shows that overall structure of isolated viruses may not be significantly affected, while loop regions connected to beta sheets are varied among the spike proteins of four variants. When predicted affinity of spike protein with ACE2

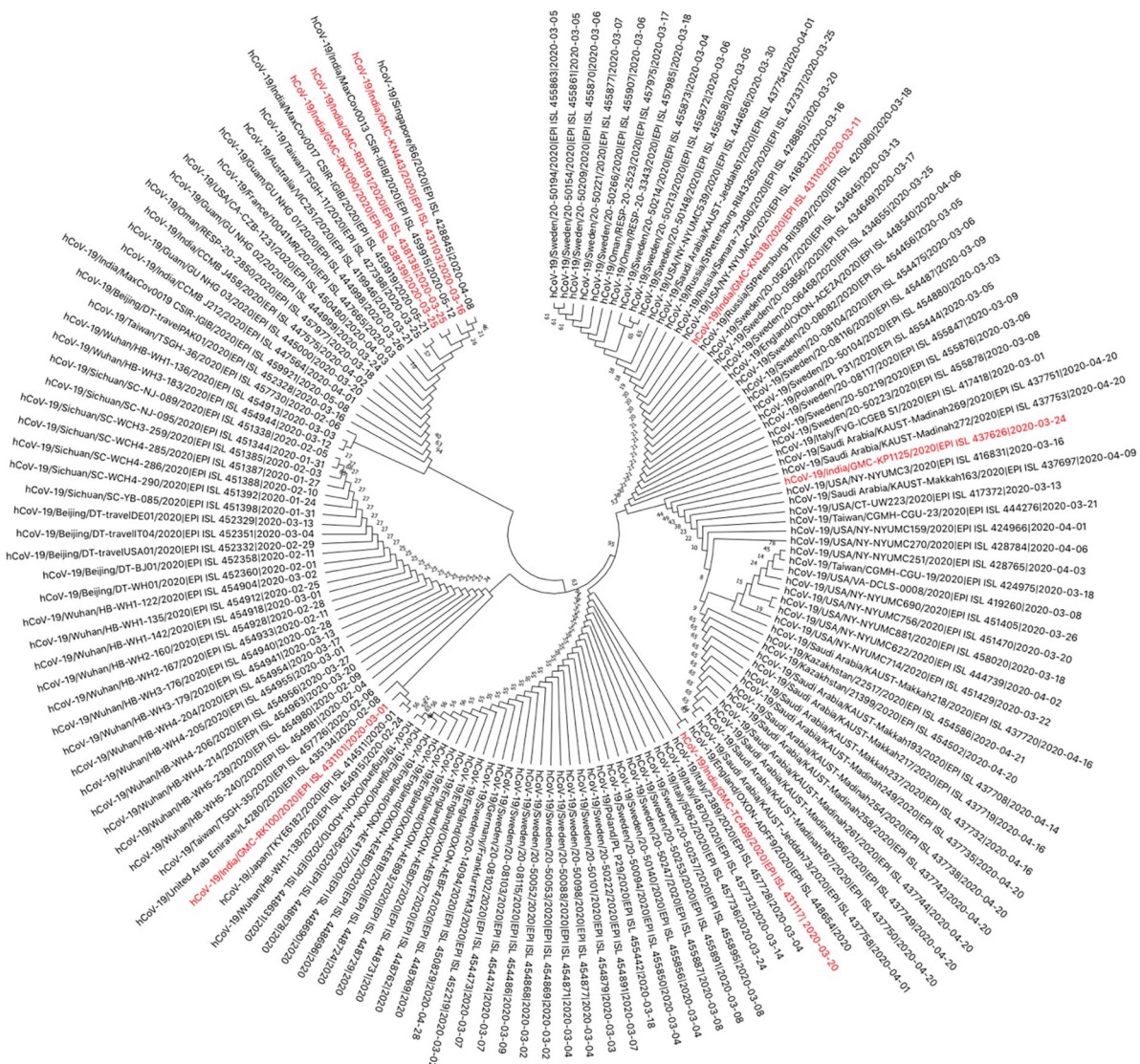

**Fig 4. Phylogenetic tree of Indian SARS-CoV-2 isolates with sequences worldwide.** Indian SARS-CoV-2 sequences of the current study are shown in red.

was analysed through docking, the results showed that spike protein of RK100 may be interacted between position 470–490 and 200–220, while KN318, TC469 and KP1125 spike protein may be interacting with ACE2 between position 250–270 and 630–650 (Fig 7). In case of KN443, RK1090 and RR1191, spike protein interacts with ACE2 between position 70–80, 120–160, 170–250 and 470–485. When energy of interaction and binding constant was compared with Wuhan reference, the results showed that the binding affinity RK100 and KN318 decreased by 5615 and 6000-fold, respectively. Thus, suggesting that the virus affinity to ACE2 receptor could be varied among the three variants of the isolates in circulation (Table 4).

We observed 3 distinct variants Y28H (RK100), D614G (KN318, TC469, KP1125), and similar to Wuhan reference NC_045512.2 (KN443, RK1090 and RR1191). These mutations were distinctly different with varied affinities as assessed by theoretical analysis, wherein the affinity of Wuhan was higher followed by Y28H and D614G (Table 4). When RNA reads for ACE2

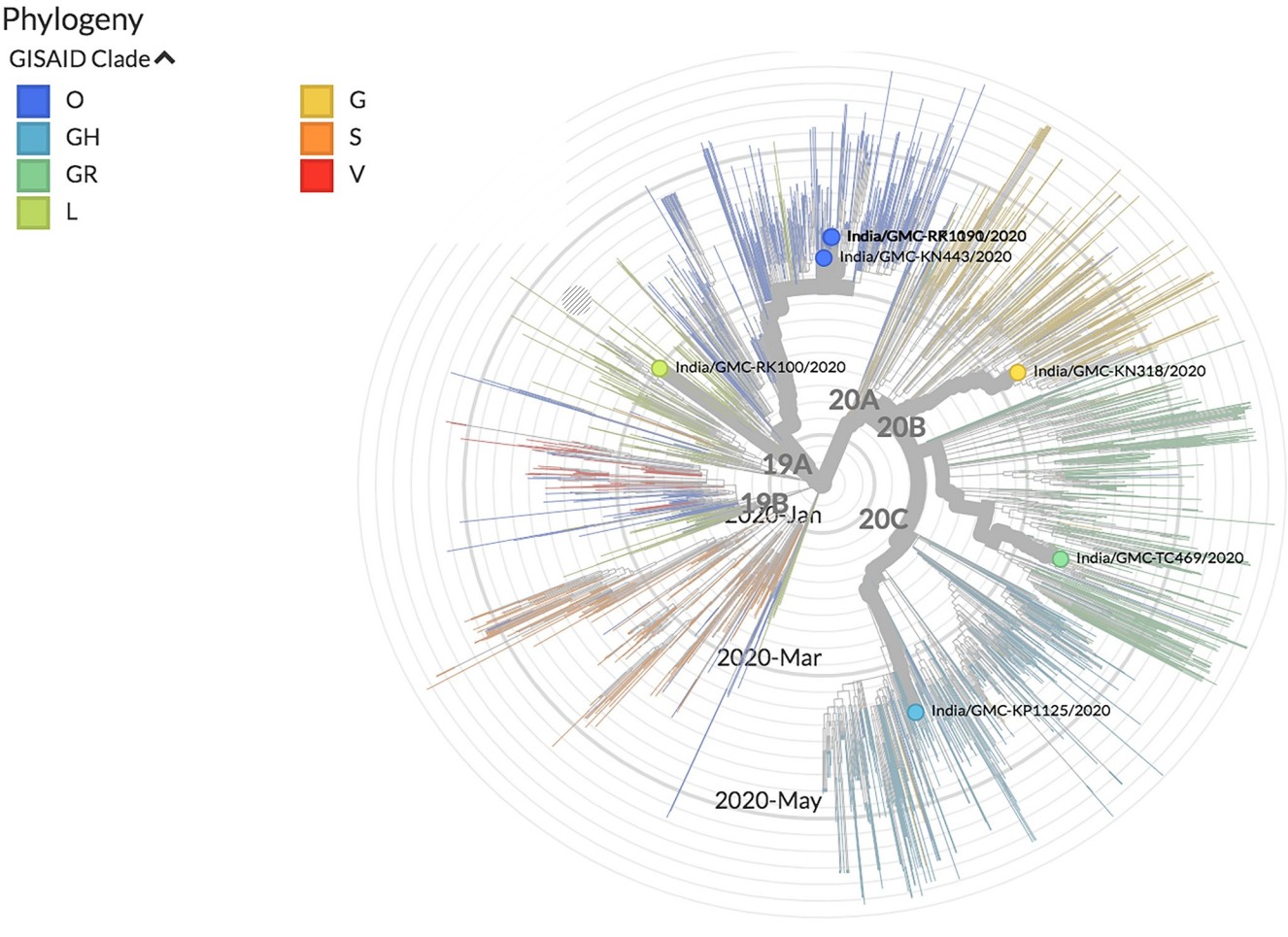

**Fig 5. The clustering of the isolates in the current study in the phylogenetic tree of all SARS-CoV-2 sequences globally according to the GISAID database.**

were quantified, it shows higher variability among the patients, but the ACE2 expression (0.13 to 1.51) (Table 4) could not be correlated with the predicted affinity of the S protein to ACE2 of isolates and also Ct values. A larger WGS sample size, which is a limitation of the current study, can provide better insights in this area. It was interesting to note comparatively low Ct values (high viral loads) in the Spike G614 variants when compared to D614 variants (E gene: 19±1 vs 22±2; RdRp: 26±0 vs 27±5; ORF: 25±1 vs 28±3).

## Conclusion

Clinical profile of COVID-19 in India majorly shows fever and cough; comorbid diabetes, hypertension, and CVD were associated with increased severity of infection. Increased neutrophil count, decreased lymphocyte count, increased levels of SGOT, SGPT and blood urea were also associated with increased severity of infection. The observations in X-ray and CT scan confirms the severity of infection. A limitation of the current study is the inability to retrieve all the data regarding radiological findings. This hindered us to compare the findings extensively in both groups. It is hoped that findings from this study will guide clinicians to identify patients with different prognosis at an early stage based on the clinical characteristics

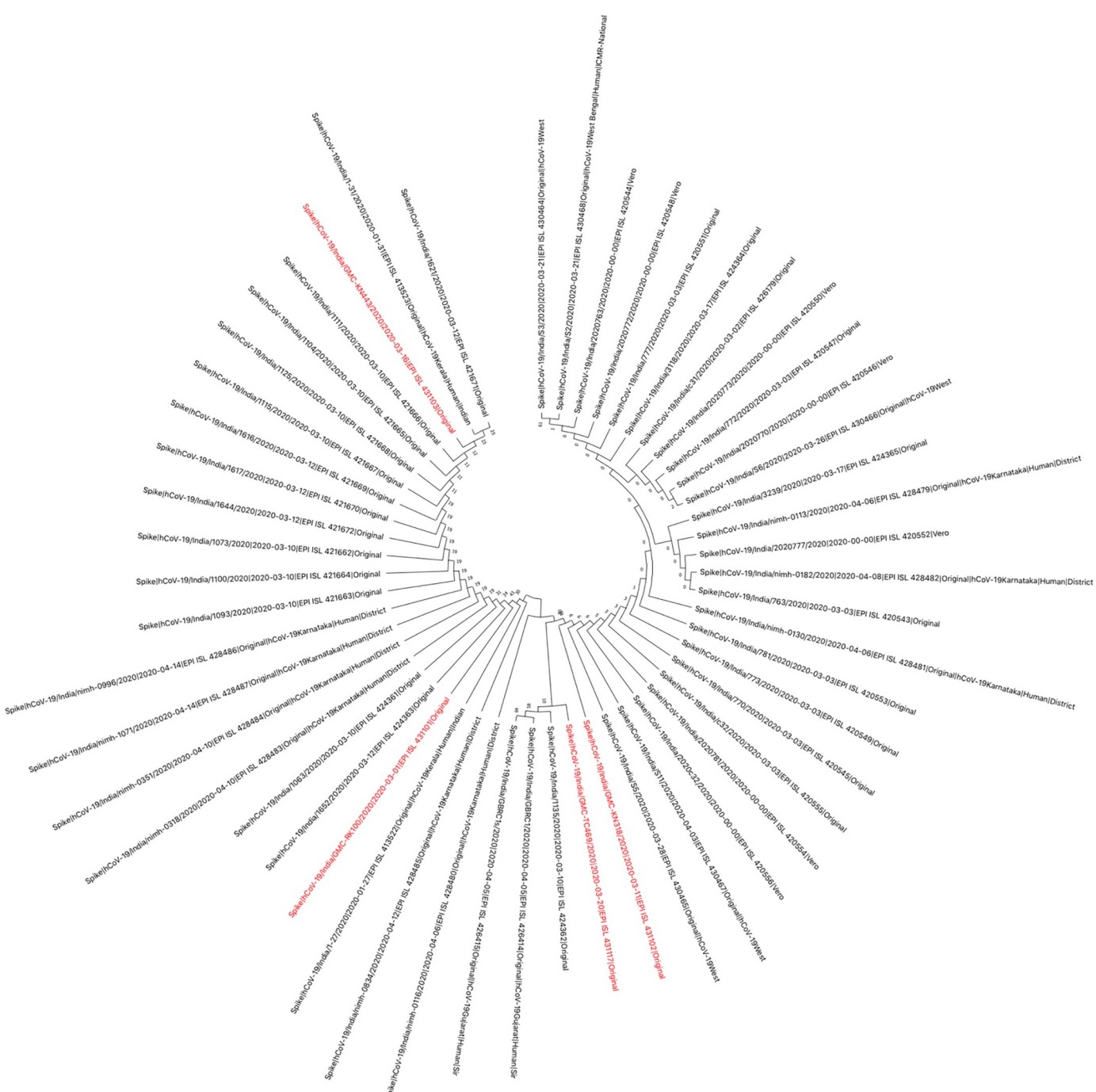

**Fig 6. The phylogenetic tree of SPIKE protein from all the seven Indian Isolates.**

presented by patients, and help in providing appropriate and effective management for the patients. Sequence analysis of seven patients of SARS CoV-2 in India showed they are under two clades I/A3i, A2a; and O, L, GR, and GH clades as per the latest GISAID classification. Analysis of the S protein showed that they belong to three variants with distinct predicted ACE2 interaction. ACE2 expression levels were variable among the patients.

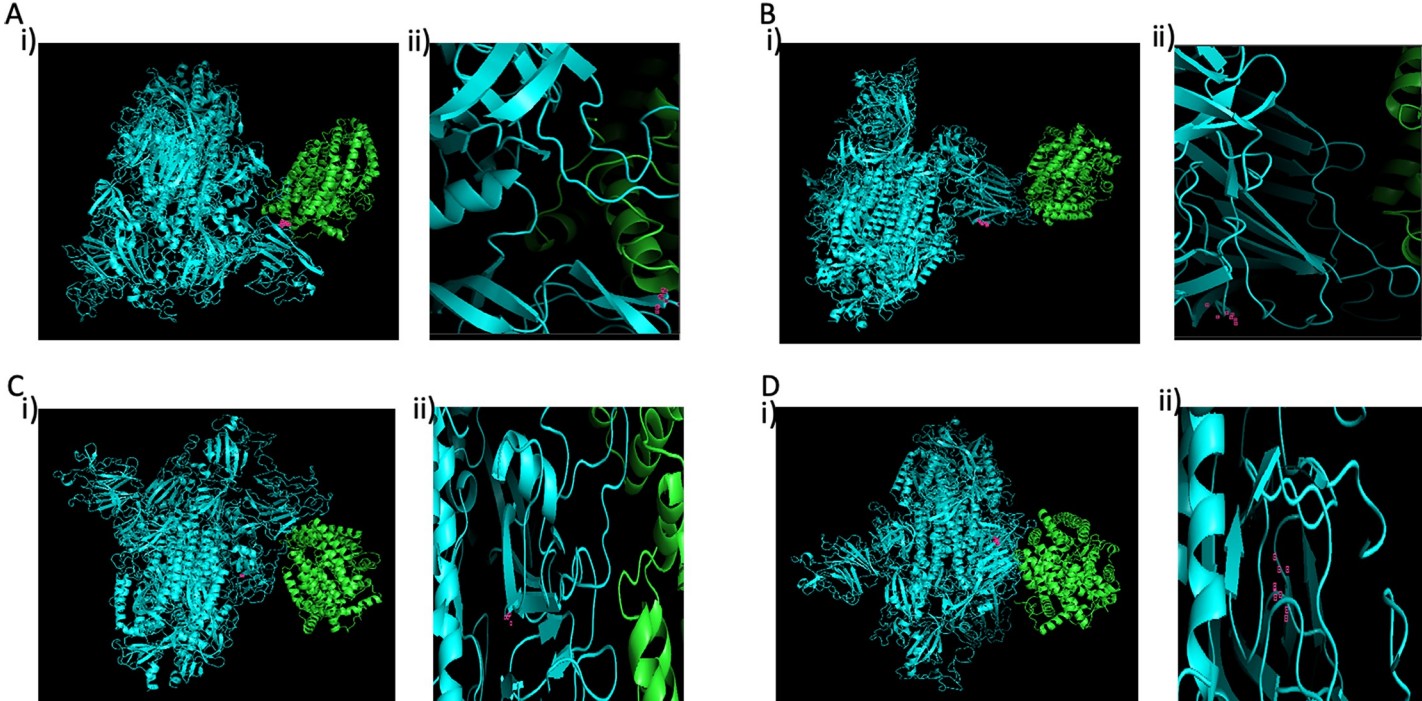

**Fig 7. Spike protein ACE2 docking.** Comparison of predicted docking origination of ACE2 interaction with (A) RK100 (Y28H) interacts with ACE2 through its residues 623 to 639 along with 293 Leu and 28 His; (B, D) KN443, RK1090 and RR1191 interacting ACE2 through its residues 152 to 184; (C) KN318, TC469 and KP1125 (D614G) interreacts ACE2 through its residues 623 to 639. 614G is located in loops. AA being referred is indicated in pink dots. Panel (ii) of 7A, 7B, 7C, and 7D indicates zoomed in image of AA side chains involved in interaction that was depicted in panel (i), respectively.

Table 4. Energy and dissociation constants of interaction of Spike protein with host ACE2.

| Docked structure with ACE2 | Binding Affinity (Kcal/mol) | Dissociation constant (M) | ACE2 Expression (FPKM) in Sample |
|---|---|---|---|
| RK100 | -50.6 | 7.8E-38 | 0.5576 |
| KN318 | -49.9 | 2.6E-37 | 4.1271 |
| TC469 | -49.9 | 2.6E-37 | 0.1397 |
| KP1125 | -49.9 | 2.6E-37 | 1.5134 |
| KN443 | -55.8 | 1.3E-41 | 0.3940 |
| RK1090 | -55.8 | 1.3E-41 | 0.4303 |
| RR1191 | -55.8 | 1.3E-41 | 0.4722 |

## Supporting information

**S1 File.**
(XLS)

## Acknowledgments

We thank Dr. Rajith Kumar, IAS, Principal Secretary, Telangana State council Science and Technology, Government of Telangana, Dr.G.Prakash Rao Principal, Gandhi Medical College and Hospital, Dr.A.Vinaya Sekhar, HOD, Department of General Medicine, GMC, Hyderabad, Prof. S. Satayanarayana, Former Vice Chancellor, Osmania University, Hyderabad for

their support, and Medgenome, Bangalore for their technical support in WGS. We are indebted to all the patients and staff from Department of Microbiology and General Medicine, Gandhi Medical College and Hospital who assisted in this study. We sincerely appreciate the researchers worldwide who sequenced and shared the complete genome data of SARS-CoV-2 and other coronaviruses from GISAID (https://www.gisaid.org/). Dr. Kondapi participated in the analysis during his sabbatical period.

## Author Contributions

**Conceptualization:** Nagamani Kammili, Thrilok Chander Bingi, Anand K. Kondapi.

**Data curation:** Radhakrishna Muttineni, Nagamani Kammili, Thrilok Chander Bingi, Raja Rao M., Pankaj Singh Dholaniya, Sunitha Pakalapati, Saritha S., Shekar K.

**Formal analysis:** Radhakrishna Muttineni, Nagamani Kammili, Thrilok Chander Bingi, Raja Rao M., Pankaj Singh Dholaniya, Sunitha Pakalapati, Mallikarjuna Reddy Doodipala, Amit A. Upadhyay, Steven E. Bosinger, Anand K. Kondapi.

**Funding acquisition:** Ravi Kumar Puli.

**Investigation:** Radhakrishna Muttineni, Thrilok Chander Bingi, Raja Rao M., Sunitha Pakalapati.

**Methodology:** Radhakrishna Muttineni, Nagamani Kammili, Thrilok Chander Bingi, Raja Rao M., Pankaj Singh Dholaniya.

**Project administration:** Nagamani Kammili, Ravi Kumar Puli.

**Supervision:** Rama R. Amara, Anand K. Kondapi.

**Validation:** Mallikarjuna Reddy Doodipala, Amit A. Upadhyay, Steven E. Bosinger.

**Visualization:** Kalyani Putty.

**Writing – original draft:** Kalyani Putty.

**Writing – review & editing:** Rama R. Amara, Anand K. Kondapi.

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
