## [Decision Letter · Decision Letter 0]

5 Nov 2020

PONE-D-20-25334

Clinical and Whole Genome Characterization of SARS-CoV-2 in India

PLOS ONE

Dear Dr. Kondapi,

Thank you for submitting your manuscript to PLOS ONE. After careful consideration, we feel that it has merit but does not fully meet PLOS ONE’s publication criteria as it currently stands. Therefore, we invite you to submit a revised version of the manuscript that addresses the points raised during the review process. One of the reviewers has pointed major changes and must be addressed in the manuscript.

We look forward to receiving your revised manuscript.

Kind regards,

Binod Kumar, PhD

Academic Editor

PLOS ONE

Journal Requirements:

2. Thank you for stating in the text of your manuscript:

"The study was carried out as per the Institutional Ethics Committee approval No. IEC/GMC/2020/02/40 dated 04/04/2020 of Gandhi Medical College and Hospital. In the

context of emerging infectious disease conditions, the requirement for written informed consent was waived."

Please also add this information to your ethics statement in the online submission form and specifically state that your study was approved by the ethics committee.

Reviewers' comments:

Reviewer's Responses to Questions

**Comments to the Author**

1. Is the manuscript technically sound, and do the data support the conclusions?

Reviewer #1: Partly

Reviewer #2: Yes

2. Has the statistical analysis been performed appropriately and rigorously? 

Reviewer #1: No

Reviewer #2: Yes

3. Have the authors made all data underlying the findings in their manuscript fully available?

Reviewer #1: Yes

Reviewer #2: Yes

4. Is the manuscript presented in an intelligible fashion and written in standard English?

Reviewer #1: No

Reviewer #2: Yes

5. Review Comments to the Author

Reviewer #1: Dear Associate Editor,

The manuscript 'Clinical and Whole Genome Characterization of SARS-CoV-2 in India’ by Kondappi et al analyze the clinical and biochemical profile and whole genome sequences of patients with SARS-CoV-2 in a tertiary hospital in India and thereby providing some guidelines that might aide in the treatment of those patients at an early stage. However, there are some concerns in this manuscript, which the authors have failed to address.

Major comment:

1. In the introduction, the authors nicely summarize about the various strains of human coronaviruses and its complications, however they have missed to cite many key literatures in the text.

a. The key reference of SARS-CoV-2 (Zhu et al, NEJM 2020) according to their statement in the first sentence is missing.

b. There are no references in the second paragraph of the introduction, which includes the statements about WHO and India. Without a proper reference, the sentence on the mitigation efforts by the Indian government looks like an overstatement, since the number of cases appear to have risen just before this manuscript was submitted.

c. Since one of the important finding in this study is focused on the comparison of different strains and mutations and the samples have been obtained from patients with international history, there should be a sentence on how this particular information was collected and their authenticity.

2. One of the aims in this study is to present the clinical difference amongst the SARS-CoV-2 patients. For this reason, the authors need to present this section more clearly and understandable. However, there is so much confusion and disparity in this section in both the text and the table.

a. It is understandable the authors are making a comparison of those patients who recovered and succumbed and draw comparison to other studies. However, this part is confusing as their classification in the text is based on the patient’s age groups and also their outcome, whereas the table is focused only on the latter.

b. In table 1, those items written within the brackets are confusing, therefore it will be better if those items in the bracket are listed for each and every item listed in the rows of the table.

c. There is a disparity in the number and the frequency for the cause of death in table 1.

d. Rather than writing ‘significant difference’, the authors should explicitly write as increased or decrease of that particular component when you compare R to S group.

3. In table 1, the authors indicate there was a significant difference between R and S group in regards to the supportive therapy. Can the authors comment and discuss on which drugs were used and what impact they had?

4. The role of increased neutrophils in SARS-CoV-2 should be discussed properly rather than just providing a statement from the reference. Similarly, the role of decreased lymphocytes in context of SARS-CoV-2 (if not in other Coronavirus infections) should also be discussed.

5. The authors do not mention anything about the ORF1ab, ORF3a, Spike, nucleocapsid or about ACE2 in the methods section and they also do not discuss about their importance or relevance to SARS-CoV-2 in this manuscript. The authors indicate the whole genome sequencing was performed only in patients with international travel history, but not in others, is there a reason as to why there is a bias in this, as they anyway compare their sequences to those available from worldwide to determine the points of transmission.

6. In the conclusion, the authors mention the patients exhibit higher immunological response compared to previous published reports. It is not clear as to what they mean by higher immunological response.

7. In the conclusion, the authors mention the radiological findings confirms the severity of infection. This is not clear, as they do not interpret the radiological findings in figure 1 as to how it is related to SARS-CoV-2. In addition to that, the authors need to indicate the number of patients for whom the radiological diagnosis were performed and their criteria, if any.

Minor comments:

1. The authors need to pay attention to the errors related to English grammar in the results section and address that.

2. The authors should also provide references in the methods section, wherever necessary, and this includes their description on sample collection by WHO guidelines and using the manufacturer/others protocol.

3. In the results section, the opening paragraph is not required, since the exact same information has been provided under the clinical and biochemical profile.

4. The authors need to pay attention to the abbreviation when a particular word is written for the first time in the text or table.

5. It would be better if the authors indicate which statistical tests were used for comparing the difference between R and S groups in table 1 along with the results.

6. Which disease does reference 27 indicate?

7. Ratios are usually indicated as 1:2 and not as a number with +, clarity should be provided on the neutrophil and lymphocyte ratio.

8. There is no reference for indicating SGOT and SGPT levels are associated with heart and liver infection.

9. There is no reference for the health bulletin provided by Telangana government.

10. In Figure 2, is the Clade – I/A3a or I/A3i, they are different in the text and figure.

11. Is there any scale to distinguish between the two different panels in figure 6 A-D?

Reviewer #2: This study is pertinent, considering the raging current pandemic. Authors of this report provided a correlation of increased risk of severe illness with underlying conditions and immunological responses in the Indian patients. This report will help in better management of patients in the clinical settings.

6. PLOS authors have the option to publish the peer review history of their article (what does this mean?). If published, this will include your full peer review and any attached files.

Reviewer #1: **Yes: **Godhev Manakkat Vijay

Reviewer #2: **Yes: **Vikram Srivastava

---

## [Author Response · Author response to Decision Letter 0]

12 Dec 2020

Response to reviewer’s comments/suggestions for manuscript entitled “Clinical and whole genome characterization of SARS-CoV-2 in India”

We thank you for considering the manuscript for publication in PLoS One. We thank the reviewers for providing constructive comments and helping with improvement of the manuscript quality. We have gone through each point carefully and the necessary answers are given below point by point.

Response to the Editorial comments:

Ans: We have followed the style template provided in PLoS One website in the revised version.

2. Thank you for stating in the text of your manuscript:

"The study was carried out as per the Institutional Ethics Committee approval No. IEC/GMC/2020/02/40 dated 04/04/2020 of Gandhi Medical College and Hospital. In the context of emerging infectious disease conditions, the requirement for written informed consent was waived." Please also add this information to your ethics statement in the online submission form and specifically state that your study was approved by the ethics committee.

Ans: As per the journal’s suggestion, the necessary details are now included in the online submission form.

a. Please clarify the sources of funding (financial or material support) for your study. List the grants or organizations that supported your study, including funding received from your institution.

Ans: The authors received no specific funding for this work.

Ans: The authors received no specific funding for this work. Hence not applicable to this manuscript.

Ans: Not applicable for the current study.

d. If you did not receive any funding for this study, please state: “The authors received no specific funding for this work.”

Ans: As per the journal’s suggestion, the necessary details are now included in line 516 of the revised manuscript.

Response to the major comments of the reviewers:

1. In the introduction, the authors nicely summarize about the various strains of human coronaviruses and its complications, however they have missed to cite many key literatures in the text.

a. The key reference of SARS-CoV-2 (Zhu et al, NEJM 2020) according to their statement in the first sentence is missing.

Ans: As per the reviewer’s suggestion, Zhu et al, NEJM 2020 is now cited as reference 1 in line 77 of the revised manuscript.

b. There are no references in the second paragraph of the introduction, which includes the statements about WHO and India. Without a proper reference, the sentence on the mitigation efforts by the Indian government looks like an overstatement, since the number of cases appear to have risen just before this manuscript was submitted.

Ans: As per the reviewer’s suggestion, WHO statistics are updated as reference 13 in lines 94 of the revised manuscript. The following statement is deleted from the revised manuscript: “Prompt intervention by Government of India and the health authorities made the spread of infection under manageable limit”.

c. Since one of the important finding in this study is focused on the comparison of different strains and mutations and the samples have been obtained from patients with international history, there should be a sentence on how this particular information was collected and their authenticity.

Ans: As per the reviewer’s suggestion, the necessary details are now included in lines 113-119 of the revised manuscript as follows: “perform comprehensive WGS analysis of seven strains of SARS-CoV-2 from among the first reported (during February-March, 2020) SARS-CoV-2 cases in Hyderabad, India with international travel history from Europe, USA, Indonesia, and United Arab Emirates. These patients were screened for COVID-19 symptoms at Rajiv Gandhi International airport, Hyderabad, India, quarantined, and were tested positive for COVID-19 by RT-PCR. These patients were admitted in the COVID referral hospital in Hyderabad and enrolled in the study.”

2. One of the aims in this study is to present the clinical difference amongst the SARS-CoV-2 patients. For this reason, the authors need to present this section more clearly and understandable. However, there is so much confusion and disparity in this section in both the text and the table.

a. It is understandable the authors are making a comparison of those patients who recovered and succumbed and draw comparison to other studies. However, this part is confusing as their classification in the text is based on the patient’s age groups and also their outcome, whereas the table is focused only on the latter.

Ans: As per the reviewer’s suggestion, to simplify and for ease of understanding, patients are now only grouped as “recovered” and “succumbed”. Since comparison of clinical and biochemical parameters was not done as per the age, the following statement is deleted from the revised manuscript: “The patients were divided into four different age groups, child (3-12 years old; N=6), adolescent (13-18 years old; N=6), adult (19-59 years old; N=62), and senior (> 60 years old; N=35)”. Also, this part of the results section is edited to be presented clearly. Please refer to lines 200-312 of the revised manuscript.

b. In table 1, those items written within the brackets are confusing, therefore it will be better if those items in the bracket are listed for each and every item listed in the rows of the table.

Ans: The numbers in the brackets indicate percentage of patients enrolled in the current study that are exhibiting the respective parameters. 

c. There is a disparity in the number and the frequency for the cause of death in table 1.

Ans: As per the reviewer’s suggestion, this was noted and now corrected.

d. Rather than writing ‘significant difference’, the authors should explicitly write as increased or decrease of that particular component when you compare R to S group.

Ans: As per the reviewer’s suggestion, the necessary details are now included in the revised manuscript.

3. In table 1, the authors indicate there was a significant difference between R and S group in regards to the supportive therapy. Can the authors comment and discuss on which drugs were used and what impact they had?

Ans: Supportive therapy was provided with antibiotics (cefotaxime, Azithromycin), antivirals (oseltamivir and lopinavir/ritonavir), hydroxy chloroquine, and steroids. Interestingly, significant difference in favor of S group was noticed with supportive therapy. This suggests that the treatment started in severely ill patients had no effect in the recovery of the patients. This also suggests that supportive therapy although statistically found significant, has no effective role in disease outcome. The same is mentioned in lines 223-228 of the revised manuscript.

4. The role of increased neutrophils in SARS-CoV-2 should be discussed properly rather than just providing a statement from the reference. Similarly, the role of decreased lymphocytes in context of SARS-CoV-2 (if not in other Coronavirus infections) should also be discussed.

Ans: Neutrophilia is a hallmark of any acute infection. It was also suggested before that aberrant activation of neutrophils contribute to the exacerbated host response in patients with severe COVID-19, predicting poor outcome. Importantly neutrophil infiltration was also observed in pulmonary capillaries of COVID-19 patients leading to acute capillaritis with fibrin deposition, extravasation of neutrophils into the alveolar space, and neutrophilic mucositis. In addition to this, the role of neutrophilia as a source of excess neutrophil extracellular traps (NETs) was also discussed. NETs are web-like structures of DNA and proteins expelled from the neutrophils that capture and kill the pathogens. In addition to being useful, excessive NET formation can trigger a cascade of inflammatory reactions that results in permanent organ damage to the pulmonary, cardiovascular, and renal systems, the most commonly affected organ systems in severe COVID-19. 

Lymphocytes play an important role in the maintenance of immune system function. After virus infection, alteration in total lymphocyte numbers varies with different viral pathogenic mechanisms. Several mechanisms were shown before for virus induced lymphopenia including immune injuries from inflammatory mediators, virus attachment, exudation of circulating lymphocytes into inflammatory lung tissues. Recent studies indicated a clear association of decrease in peripheral lymphocytes with the clinical characteristics of COVID-19 patients. This indicates an impairment of the immune system during the course of SARS-CoV-2 infection. These alterations were also found in patients with pneumonia caused by MERS- CoV and SARS-CoV. 

As per the reviewer’s suggestion, the necessary details are now included in lines 258-269, and 278-287 of the revised manuscript.

5. The authors do not mention anything about the ORF1ab, ORF3a, Spike, nucleocapsid or about ACE2 in the methods section and they also do not discuss about their importance or relevance to SARS-CoV-2 in this manuscript. The authors indicate the whole genome sequencing was performed only in patients with international travel history, but not in others. Is there a reason as to why there is a bias in this, as they anyway compare their sequences to those available from worldwide to determine the points of transmission.

Ans: The mutations observed in genes other than Spike are discussed (lines 363-378) in manuscript as per the reviewer’s suggestion. The discussion for ORF1ab and N protein is incorporated while there was no mutation observed in ORF3a. As a part of this discussion, a new figure (Fig 2) is now added to the revised manuscript depicting the graphical representation of polyprotein synthesized by orf1ab. The earlier mention of the mutation 24130C>T in ORF3a under the heading “Genetic variation in the sequences” in “Results and Discussion” section was incorrectly written. This position doesn’t lie in ORF3a, but belongs to the S gene. The text in the manuscript is now been corrected (line 335). As we did not analyze the mutation in ACE2 gene among different patients, discussion about ACE2 is out of scope of this manuscript, however the binding affinities of human ACE2 protein with the Spike protein has been studied and discussed in the manuscript.

The samples for the genome sequencing were taken from the patients with the international travel history, as they were the first samples available at the time of the study being initiated. No specific reason is considered in collecting the samples from patients with the international travel history. The comparison with globally available sequences is performed to compare the distribution of the samples among worldwide based on similarity and not to determine the point of transmission. 

6. In the conclusion, the authors mention the patients exhibit higher immunological response compared to previous published reports. It is not clear as to what they mean by higher immunological response.

Ans: As per the reviewer’s suggestion, the sentence is removed from the revised manuscript due to the generalised and vague nature of the statement. 

7. In the conclusion, the authors mention the radiological findings confirms the severity of infection. This is not clear, as they do not interpret the radiological findings in figure 1 as to how it is related to SARS-CoV-2. In addition to that, the authors need to indicate the number of patients for whom the radiological diagnosis were performed and their criteria, if any.

Ans: Radiological results are now discussed in lines 250-256 of the revised manuscript. A limitation of the current study is being unable to retrieve all the data regarding radiological findings which hindered us to compare the findings extensively in both groups. The same has been mentioned in lines 505-507 of the revised manuscript.

Response to the minor comments of the reviewers:

1. The authors need to pay attention to the errors related to English grammar in the results section and address that.

Ans: As per the reviewer’s suggestion, the manuscript is corrected with respect to English grammar to the best of our abilities.

2. The authors should also provide references in the methods section, wherever necessary, and this includes their description on sample collection by WHO guidelines and using the manufacturer/others protocol.

Ans: As per the reviewer’s suggestion, reference numbers 13, 14, 15 are now included in methods section on sample collection.

3. In the results section, the opening paragraph is not required, since the exact same information has been provided under the clinical and biochemical profile.

Ans: As per the reviewer’s suggestion, the opening paragraph of the results is now removed in the revised manuscript to avoid repetition.

4. The authors need to pay attention to the abbreviation when a particular word is written for the first time in the text or table.

Ans: Noted, and done.

5. It would be better if the authors indicate which statistical tests were used for comparing the difference between R and S groups in table 1 along with the results.

Ans: The statistical tests used in the current study for comparing the clinical findings is explained in the methods section in lines 189-198.

6. Which disease does reference 27 indicate?

Ans: Reference 27 is now reference 28 in the revised manuscript, and it refers to COVID-19.

7. Ratios are usually indicated as 1:2 and not as a number with +, clarity should be provided on the neutrophil and lymphocyte ratio.

Ans: Kindly note that there is no reference to ratios in the revised manuscript.

8. There is no reference for indicating SGOT and SGPT levels are associated with heart and liver infection.

Ans: As per the reviewer’s suggestion, reference 43 is now included for the same, in line 295 of the revised manuscript.

9. There is no reference for the health bulletin provided by Telangana government.

Ans: As per the reviewer’s suggestion, references 45, 46 are now included for the same, in line 307 of the revised manuscript.

10. In Figure 2, is the Clade – I/A3a or I/A3i, they are different in the text and figure.

Ans: In the text (under the heading “Phylogenetic analyses” in “Results and Discussion”) it was mistakenly written as I/A3a, which is now corrected to I/A3i in line 408 of the revised manuscript.

11. Is there any scale to distinguish between the two different panels in figure 6 A-D?

Ans: There is no scale to distinguish between the two different panels in figure 6 A-D (now figure 7). Only the zoomed picture of an interaction is represented for better visibility and convenience without following a fixed scale.

---

## [Decision Letter · Decision Letter 1]

15 Jan 2021

Clinical and Whole Genome Characterization of SARS-CoV-2 in India

PONE-D-20-25334R1

Dear Dr. Kondapi,

We’re pleased to inform you that your manuscript has been judged scientifically suitable for publication and will be formally accepted for publication once it meets all outstanding technical requirements.

Kind regards,

Binod Kumar, PhD

Academic Editor

PLOS ONE

Additional Editor Comments (optional):

Reviewers' comments:

Reviewer's Responses to Questions

**Comments to the Author**

1. If the authors have adequately addressed your comments raised in a previous round of review and you feel that this manuscript is now acceptable for publication, you may indicate that here to bypass the “Comments to the Author” section, enter your conflict of interest statement in the “Confidential to Editor” section, and submit your "Accept" recommendation.

Reviewer #1: All comments have been addressed

2. Is the manuscript technically sound, and do the data support the conclusions?

Reviewer #1: Yes

3. Has the statistical analysis been performed appropriately and rigorously? 

Reviewer #1: Yes

4. Have the authors made all data underlying the findings in their manuscript fully available?

Reviewer #1: Yes

5. Is the manuscript presented in an intelligible fashion and written in standard English?

Reviewer #1: Yes

6. Review Comments to the Author

Reviewer #1: All have comments have been addressed.

Additional suggestion: It would be better if the abbreviations R and S are changed to Recovered and Succumbed in the text, as it will be easier for the readers (this is because R and S are not universal abbreviations).

This is not mandatory, this change be made before submitting the final version.

7. PLOS authors have the option to publish the peer review history of their article (what does this mean?). If published, this will include your full peer review and any attached files.

Reviewer #1: **Yes: **Godhev Kumar Manakkat Vijay

---

## [Editor Report · Acceptance letter]

22 Jan 2021

PONE-D-20-25334R1 

Clinical and whole genome characterization of SARS-CoV-2 in India 

Dear Dr. Kondapi:

I'm pleased to inform you that your manuscript has been deemed suitable for publication in PLOS ONE. Congratulations! Your manuscript is now with our production department. 

Kind regards, 

on behalf of

Dr. Binod Kumar 

Academic Editor

PLOS ONE